# Geostatistical analysis and mapping of malaria risk in children of Mozambique

**Bedilu Alamirie Ejigu** [ID] *

Department of Statistics, College of Natural and Computational Sciences, Addis Ababa University, Addis Ababa, Ethiopia

* bedilu.alamirie@aau.edu.et

## Abstract

Malaria remains one of the most prevalent infectious diseases in the tropics and subtropics, and Mozambique is not an exception. To design geographically targeted and effective intervention mechanisms of malaria, an up-to-date map that shows the spatial distribution of malaria is needed. This study analyzed 2018 Mozambique Malaria Indicator Survey using geostatistical methods to: i) explore individual, household, and community-level determinants of malaria in under-five children, ii) prepare a malaria prevalence map in Mozambique, and iii) produce prediction prevalence maps and exceedence probability across the country. The results show the overall weighted prevalence of malaria was 38.9% (N = 4347, with 95% CI: 36.9%–40.8%). Across different provinces of Mozambique, the prevalence of malaria ranges from 1% in Maputo city to 57.3% in Cabo Delgado province. Malaria prevalence was found to be higher in rural areas, increased with child's age, and decreased with household wealth index and mother's level of education. Given the high prevalence of childhood malaria observed in Mozambique there is an urgent need for effective public health interventions in malaria hot spot areas. The household determinants of malaria infection that are identified in this study as well as the maps of parasitaemia risk could be used by malaria control program implementers to define priority intervention areas.

## Background

Malaria is an infectious disease caused by a parasite that is transmitted from one subject to another by blood-sucking female anopheles mosquitoes. It is a major public health problem, especially in Africa and Asia. Many countries made an incredible progress in the fight against malaria, and as a result malaria deaths have fallen by more than 50% globally between 2000 and 2015. Seventeen countries eliminated malaria, and six were certified by WHO as malaria-free [1]. Based on WHO recent report, on a global scale, progress has levelled off but no gains were achieved in reducing malaria case incidence over the last five years [2]. The WHO strategic advisory group predicts that there will be 11 million cases of malaria in Africa by 2050 [2]. The current COVID-19 pandemic places extra burden on health systems worldwide, and especially in sub-Saharan Africa which accounts for more than 90% of global malaria cases and deaths. Recently, WHO recommends, countries to move quickly to save lives from malaria in sub-Saharan Africa [1].

**Data Availability Statement:** The data we used which is the '2018 Mozambique Malaria Indicator Survey' were obtained from the DHS program (www.dhsprogram.com), but the 'Dataset Terms of Use' do not permit us to distribute this data as per data access instructions (http://dhsprogram.com/

data/Access-Instructions.cfm). To get access for the dataset you must first be a registered user of the website (www.dhsprogram.com), and download the 2018 Mozambique Malaria Indicator Survey. Interested scholars will have access to the relevant data used in this study in the same manner as it was accessed by the authors of this study from the aforementioned website.

**Funding:** The author received no specific funding for this work.

**Competing interests:** The authors have declared that no competing interests exist.

In Africa, more than two-thirds of all malaria deaths occur in children under-five years old [3]. Because of a continual fight against malaria by intervention programs, malaria infection prevalence and clinical incidence decreased by 50% and 40%, between 2000 and 2015, respectively [4]. In 2017, an estimated 219 million cases and 435 thousand deaths of malaria occurred worldwide, of which 200 million (92%) malaria cases were in the WHO African Region [3]. Fifteen sub-Saharan Africa countries and India carried 80% of the global malaria burden and Mozambique accounts for 5%. Children under 5 years of age are the most vulnerable group affected by malaria and they accounted for 61% of all malaria deaths worldwide [3].

In Mozambique, malaria is a common disease with seasonal fluctuation in all parts of the country with seasonal peak ranging from December to April [5]. The prevalence varies across different ecological zones and transmission occurs year-round with relatively higher prevalence in the northern part of the country. Various factors influence the dynamics of malaria transmission and infection ranging from natural (i.e. rainfall, temperature) to social factors. Previously conducted national surveys showed that the prevalence of malaria in under five children was 39% and 40% in the years 2011 and 2015, respectively. Malaria accounts for 29% of all deaths and 42% of deaths in under-five children in the country [6].

Even though, Mozambique's entire population is at risk of malaria due to different environmental and ecological factors [7], pregnant women and under-five children are at higher risk of severe illness due to their low immunity [8–10]. Thus, efficient interventions and preventive measures could be improved by advancing our understanding of the spatial patterns of malaria prevalence distribution and the underlying factors.

Long-lasting insecticidal nets (LLINs) is one of the main interventions mechanisms for preventing malaria infection, and the 2017-2022 Mozambique strategic plan aims to achieve one net for every two people LLINs coverage across the country [6]. Currently, disruptions to insecticide-treated net campaigns due to COVID-19 pandemic and in access to antimalarial medicines could lead to increase in the number of malaria deaths and co-morbidities. This requires malaria affected countries like Mozambique should identify malaria hot-spot areas and move quickly to save lives from malaria. The spatial distribution risk map of malaria is an important tool for effective planning, malaria control intervention, resource mobilization, monitoring and evaluation process. As a result, to advance intervention mechanisms, spatial distribution maps of malaria prevalence across the study area have been produced using geostatistical modeling approaches [11–21] with the aim of identifying areas where greatest control effort should be focused. Previously generated maps depicted the geographical distribution of malaria risk in Mozambique either at province or continental-scale [4, 7, 22–25], not at country-level. However, those maps may not reflect the current malaria situation in the country, as they rely on historical and outdated survey data. Malaria risk maps based on historical data cannot reflect the current situation which is changing due to ongoing interventions.

To date, to the best of my knowledge, the only map available for the spatial distribution of malaria prevalence in Mozambique based on recent data was produced for Chimoio [7] and Maputo province [5, 22] which do not represent the current situation since it does not take into account contemporary effects of interventions and socio-economic status. Further, recently produced reports related with malaria in Mozambique do not address the spatial distribution of malaria across survey clusters of the country by taking into account other factors [6, 26].

In this study, the 2018 Mozambique Malaria Indicator Survey (MMIS) data were analyzed using geostatistical modeling approaches to: i) identify determinant factors associated with malaria risk, ii) produce a prevalence map of malaria among children under the age of five across survey clusters and regions of Mozambique, and iii) produce prediction prevalence maps and exceedence probability across the country. Predicted malaria prevalence maps

generated in this study would help policy makers to identify high-risk areas and design targeted interventions.

## Materials and methods

### Mozambique MIS data

The data for this study were obtained from the 2018 Mozambique Malaria Indicator Survey (MMIS). The main aim of this survey was to obtain population-based estimates of malaria indicators by considering a nationally representative dataset which serves as input for strategic planning and evaluation of malaria control program([26]). Stratified two-stage sampling technique was used to select enumeration areas (EAs) and households. Sampling procedures of the survey have been mentioned in the survey final report [26]. Permission to use the 2018 MMIS data was obtained from the DHS website (www.dhsprogram.com). Fig 1 (left panel) presents the map of survey cluster locations where raw dataset were collected, and prevalence is depicted (right panel).

The geostatistical modeling includes: individual-level variables: child age, gender, anaemia level, child slept under bed net; household-level variables including educational level of the mother, household wealth index, and availability of bed nets in the household; and community-level variables: place of residence, mean temperature, estimated malaria incidence in the cluster, ITN coverage, and region were used in the analysis. Cluster-level geospatial data used in this study (ITN-coverage, malaria incidence) were obtained from DHS Program Geospatial Covariate Datasets (www.dhsprogram.com), and the construction procedures of geospatial data were explained in [27]. Furthermore, to see the change in the prevalence of malaria by different factors in the past seven years, the 2011 Mozambique demographic and health survey used [28]. In addition to this, to compare the overall risk of malaria with neighboring countries, nationwide malaria prevalence in Tanzania, Malawi, Zambia, and Zimbabwe considered.

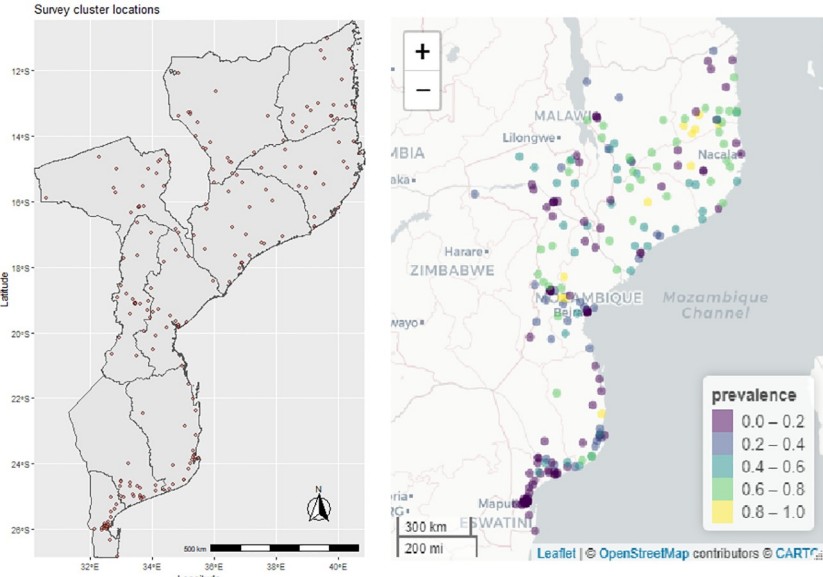

**Fig 1. Study clusters of the 2018 MMIS (left), and cluster level RDT prevalence (right panel).**

## Statistical analysis

Malaria indicator survey data-sets are often complex in nature for two reasons: i) the use of stratified multistage cluster sampling to increase sampling and cost efficiency, and ii) unequal probabilities of selection from target-populations for sampled elements, often as a result of oversampling of key subgroups. Thus, the data analysis tools employed sampling weights for generating unbiased population estimates [29].

**Weighted confidence interval for proportions.** Constructing a confidence interval for proportion $p$ is one of the most basic analyses in statistical inference, and it is an important aspect of reporting statistical results. Let $Y$ denote a binomial variate for sample size $n$, and let $\hat{p} = \frac{Y}{n}$ denote the sample proportion. Under asymptotic normality of the sample proportion and estimating the standard error, an approximate $100(1 - \alpha)$% confidence interval for $P$ is

$$\hat{p} \mp z_{1-\alpha/2} \sqrt{\hat{p}(1-\hat{p})/n}. \tag{1}$$

The large-sample Wald intervals (Eq 1) are known to perform poorly [30], but the Wilson intervals [31] given by Eq 2 have been shown to perform well in a variety of situations.

$$\frac{\hat{p} + z_{1-\alpha/2}^2/2n}{1 + z_{1-\alpha/2}^2/n} \mp \frac{z_{1-\alpha/2}\sqrt{\hat{p}(1-\hat{p}) + z_{1-\alpha/2}^2/4n}}{\sqrt{n}(1 + z_{1-\alpha/2}^2/n)} \tag{2}$$

Both Wald (Eq 1) and Wilson (Eq 2) intervals are appropriate for survey data with simple random sampling designs, but they are not designed to accommodate clustering or unequal weights of more complex sample surveys, like the data analyzed in this manuscript.

A common approach to construct confidence intervals for proportions from complex sample survey data is to modify the inputs of Eq 2 to account for survey weights and the design effect. The survey-weighted, estimated proportion, $\hat{p}$, is used along with a consistent design-based estimate, $\hat{Var}(\hat{p})$, of its variance. For complex survey data. [32] propose a modified version by replacing $z_{1-\alpha/2}$ with $t_c(1 - \alpha/2)$ in equation Eq 2, and replacing $n$ with the effective sample size, defined as $n_{eff} = n/\hat{Deff}(\hat{p})$ where

$$\hat{Deff}(\hat{p}) = \frac{\hat{Var}(\hat{p})}{\hat{p}(1-\hat{p})/n} = \frac{\sum_h \left(\frac{N_h}{N}\right)^2 \left(1 - \frac{n_h}{N}\right) \frac{\hat{p}(1-\hat{p})}{n_h}}{\hat{p}(1-\hat{p})/n} = \frac{n \sum_i^n w_i^2}{\left(\sum_i^n w_i\right)^2}$$

as the estimated design effect ([33]), $w_i$ is the weight of the $i^{th}$ unit selected in the sample, $w$ represents sampling weights that denote the inverse of the probability that the observation is included because of the sampling design.

**Geostatistical modeling.** Non-spatial modelling approaches assume independence between study locations where the data collected inadvertently neglect potential spatial dependency between neighboring locations due to unobserved common exposures. To overcome such limitations, geostatistical models relate disease prevalence data with potential predictors and quantify spatial dependence via the covariance matrix of a Gaussian process facilitated by adding random effects at the observed locations [34]. Such type of geostatistical models have already been applied to model malaria risk at different geographical scales in different Africa countries [11–13, 15, 19, 20, 35].

In model Eq 3 below, malaria status $Y_{ij}$ of child $i$ at location $j$ take a value of 1 if the child has malaria and 0 otherwise; follows a Bernoulli probability distribution. Conditionally on a zero-mean stationary Gaussian process $S(l)$ and additional set of study location specific

random effects $b_j$, the linear predictor of the model assumes the form:

$$log\left(\frac{p_{ij}}{1-p_{ij}}\right) = x'_{ij}\beta + S(l_j) + b_j. \tag{3}$$

In Eq 3 $x$ is a vector of child, household, and cluster-level covariates with associated regression coefficients $\beta$, $S = (S(l): l \in R^2)$ is a Gaussian process with mean zero, and variance $\sigma^2$, and correlation function $\rho(u) = Corr(S(l, S(l'))$. Among different parametric families, such as exponential, Gaussian, spherical have been proposed for $\rho(u)$. [36] recommends the use of Matern correlation function [37] given by

$$\rho(u; \phi, \kappa) = \frac{2^{\kappa-1}\Gamma(\kappa)}{(u/\phi)^{\kappa}\kappa_{\kappa}(u/\phi)}, u > 0$$

where $\phi > 0$ is a scale parameter, $\kappa > 0$ is the shape parameter, $\kappa_{\kappa}(\cdot)$ is the modified Bessel function of the second kind of order $\kappa > 0$, and $u = ||l - l'||$ is the Euclidean distance between two locations.

In Eq 3, location-specific random effects $b_j$ were included in the model to account for unexplained non-spatial variation. These random effects are assumed to be independent normal distributed with zero mean and variance $\tau^2$ (i.e. $b_j \sim N(0, \tau^2)$), with $\tau^2$ as the nugget effect accounting for the non-spatial variation. The marginal distribution of the outcome variable in Eq 3 is a multivariate Gaussian process with mean vector $X\beta$ and covariance matrix $\Sigma(\theta)$ with diagonal elements $\sigma^2 + \tau^2$ and off-diagonal elements are $\sigma^2 \rho(u)$, with $u$ the distance between locations $l$ and $l'$.

Since Monte Carlo methods enable flexibility in fitting complex models and minimize computational problems encountered in the solely likelihood-based fitting [38, 39], in this study the model fitting was carried out using Monte Carlo maximum likelihood, as opposed to MCMC methods by considering the PrevMap package in R [40]. The likelihood function for parameters $\beta$ and $\theta^T = (\sigma^2, \phi, \tau^2)$ is obtained by integrating out the random effects included in Eq 3, where $\sigma^2$ is the variance, $\phi$ is the range, and $\tau^2$ is the nugget effect. Furthermore, to identify different risk factors, by taking in to account survey design weights, a non-spatial multilevel mixed model is fitted to the data and results presented in the S1 File.

**Spatial prediction.** For mapping, we predicted prevalence of infection at 7892 grid locations covering the entire Mozambique. Since it is difficult to get individual-level data at prediction location, The predictive map of malaria risk in Mozambique was created using the null geostatistical model 3.

Spatial distribution maps of malaria prevalence by survey clusters and regions of the country, and likelihood-based geostatistical modeling and spatial prediction were developed using R [41].

## Results

A total of 4,347 children of age 6-59 months were tested for malaria from 221 nationally representative survey clusters. Table 1 presents the overall weighted proportion of children age 6-59 months classified as having malaria based on rapid diagnostic test(RDT) results according to different background characteristics. The mean age of children in this study was 32 months with standard deviation 15.5 month, and 64.57% of the children lived in rural areas. The overall weighted prevalence of malaria by RDT in Mozambique was 38.9% (with 95% CI: 36.9%—40.8%).

According to their place of residence, the prevalence of malaria was 46.4% (with 95% CI: 43.9%–48.9%) in rural areas, and 18.5% (95% CI:16.2%-21.1%) in urban areas (Table 1). Fig 1

**Table 1. RDT prevalence of children of age 6-59 months classified as having malaria by different background characteristics and relative change between 2011 DHS and 2018 MMIS.**

| Factors | 2011 DHS | | | 2018 MIS | | |
|---|---|---|---|---|---|---|
| | n | Proportion | 95% CI | n | Proportion | 95% CI |
| **Residence** | | | | | | |
| Urban | 1599 | 0.167 | (0.146,0.191) | 1540 | 0.185 | (0.162, 0.211) |
| Rural | 3317 | 0.537 | (0.442,0.483) | 2807 | 0.464 | (0.439, 0.489) |
| **Province** | | | | | | |
| Niassa | 434 | 0.521 | (0.426,0.531) | 446 | 0.483 | (0.427, 0.539) |
| Cabo Delgado | 413 | 0.472 | (0.476,0.579) | 398 | 0.573 | (0.519, 0.625) |
| Nampula | 411 | 0.433 | (0.512,0.620) | 434 | 0.478 | (0.428, 0.529) |
| Zambezia | 594 | 0.548 | (0.409, 0.495) | 457 | 0.445 | (0.387, 0.505) |
| Tete | 455 | 0.364 | (0.583,0.684) | 373 | 0.295 | (0.245, 0.351) |
| Manica | 480 | 0.279 | (0.676,0.760) | 510 | 0.473 | (0.424, 0.522) |
| Sofala | 645 | 0.305 | (0.655,0.732) | 487 | 0.293 | (0.254, 0.336) |
| Inhambane | 342 | 0.365 | (0.578,0.689) | 352 | 0.351 | (0.300, 0.405) |
| Gaza | 404 | 0.215 | (0.740,0.823) | 374 | 0.169 | (0.129, 0.217) |
| Maputo Province | 388 | 0.032 | (0.944,0.982) | 297 | 0.012 | (0.004, 0.035) |
| Maputo City | 350 | 0.0145 | (0.965,0.994) | 219 | 0.009 | (0.002, 0.036) |
| **Mother education** | | | | | | |
| No education | 1538 | 0.467 | (0.437,0.496) | 908.00 | 0.52 | (0.483, 0.562) |
| Primary | 2261 | 0.383 | (0.359,0.408) | 1960 | 0.423 | (0.395, 0.452) |
| Secondary/higher | 1117 | 0.228 | (0.199,0.261) | 858 | 0.152 | (0.114, 0.199) |
| **Child age(years** | | | | | | |
| 1 | 565 | 0.25 | (0.208,0.297) | 514 | 0.33 | (0.277, 0.388) |
| 2 | 1156 | 0.396 | (0.362,0.431) | 923 | 0.367 | (0.326, 0.411) |
| 3 | 1076 | 0.368 | (0.334,0.403) | 1000 | 0.429 | (0.391, 0.470) |
| 4 | 1089 | 0.416 | (0.381,0.451) | 933 | 0.379 | (0.338, 0.422) |
| 5 | 1030 | 0.414 | (0.378,0.451) | 977 | 0.414 | (0.373, 0.456) |
| **Child sex** | | | | | | |
| Male | 2406 | 0.394 | (0.371,0.417) | 2154 | 0.397 | (0.368, 0.425) |
| Female | 2510 | 0.368 | (0.346,0.392) | 2193 | 0.381 | (0.354, 0.408) |
| **HH Mosquito bednet** | | | | | | |
| No | 1751 | 0.403 | (0.375,0.431) | 388 | 0.483 | (0.421, 0.620) |
| Yes | 3165 | 0.369 | (0.348,0.389) | 3959 | 0.379 | (0.359, 0.399) |
| **Wealth index** | | | | | | |
| Poorest | 874 | 0.548 | (0.510,0.586) | 852 | 0.58 | (0.539, 0.620) |
| Poor | 937 | 0.515 | (0.478,0.553) | 865 | 0.517 | (0.476, 0.559) |
| Middle | 955 | 0.412 | (0.376,0.450) | 815 | 0.422 | (0.375, 0.469) |
| Rich | 1100 | 0.257 | (0.227,0.290) | 988 | 0.207 | (0.177, 0.241) |
| Richest | 1050 | 0.054 | (0.040,0.074) | 827 | 0.028 | (0.017, 0.047) |
| **Anaemia level** | | | | | | |
| Non-anaemic | 1699 | 0.478 | (0.148,0.193) | 974 | 0.218 | (0.181, 0.259) |
| Anaemic | 3217 | 0.169 | (0.457,0.498) | 3369 | 0.434 | (0.412, 0.456) |
| **Total/National Level** | **4916** | **0.381** | **(0.365,0.397)** | 4347 | 0.389 | (0.369,0.408) |

**Table 2. Prevalence of malaria in children under five years from recent surveys in neighboring countries of Mozambique.**

| Neighboring country | Survey year & data source | Child malaria prevalence(%) |
|---|---|---|
| Tanzania | MIS 2017 | 7.312 |
| Malawi | MIS 2017 | 15.203 |
| Zambia | MIS 2018 | 9.011 |
| Zimbabwe | MIS 2016 | 0.202 |

(right panel) presents the contemporary malaria situation in survey locations of the country and can be used for malaria interventions in Mozambique.

Compared with the neighboring countries, the prevalence of malaria in Mozambique was more than twofold higher (Table 2).

The prevalence of malaria varies from province to province: lowest in Maputo (1%) and higher in Cabo Delgadi (57.3%) provinces of the country (Table 1, Fig 2, left panel).

The results of geostatistical model which took into account the spatial correlation and non-spatial multilevel analysis by including survey design weights are given in Table 3.

From the geostatistical model, place of residence, mothers educational level, child age, household wealth index, child anaemia level, cluster-level malaria incidence rate and ITN-coverage were found significantly associated with malaria infection. In the geostatistical modeling part an exponential correlation function was assumed. The assumption proved to be correct as the correlation function is supported by the empirical variogram (Fig 3).

For a child living in a rural area, the odds of being malaria parasitaemia is 2.9 times as large as the odds of a child living in urban areas. The odds of being malaria-positive for anaemic child is 3.5 times that of non-anaemic child. Further, the odds of being malaria-positive for a child whose mother education level is secondary or higher is 0.64 times less likely compared with a child whose mothers are not educated.

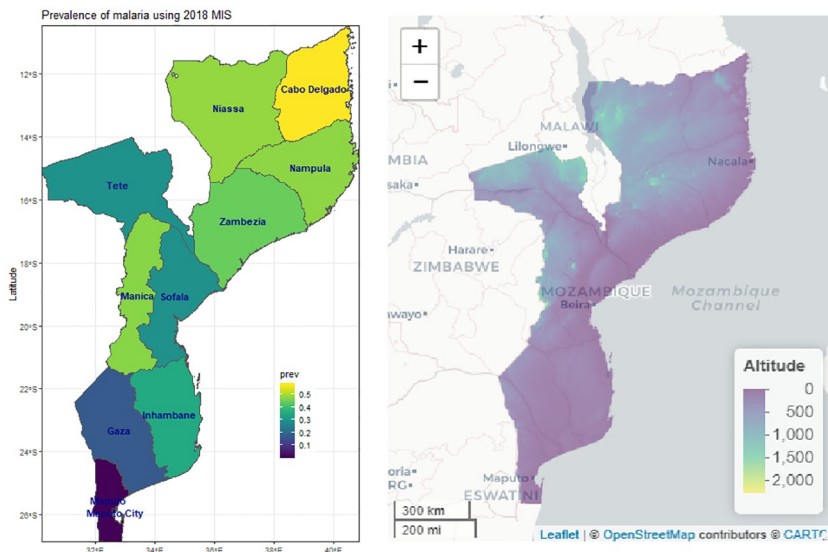

**Fig 2. Observed malaria prevalence at different provinces of Mozambique (left panel) and country altitude (right panel).**

**Table 3. Parameter estimates from the geostatistical model (3) of malaria prevalence in children under five years of age in Mozambique.**

| Factors | AOR | 95% AOR CI |
|---|---|---|
| $\hat{\beta}_0$ | 0.366 | (0.195,0.684) |
| Child level factors | | |
| Age (in moth) | 1.016 | (1.010,1.021) |
| **Sleep under bed net** (ref: No) | 0.736 | (0.557,0.971) |
| **Anemia** (ref:not anemic) | 3.502 | (2.809,4.367) |
| Household level factors | | |
| **Wealth index**(ref:Poorest) | | |
| Poor | 0.999 | (0.801,1.246) |
| Middle | 0.674 | (0.534,0.852) |
| Rich | 0.523 | (0.396,0.691) |
| Richest | 0.186 | (0.111,0.313) |
| **Mother education** (ref:No education) | | |
| Primary | 0.868 | (0.712,1.053) |
| Secondary/higher | 0.635 | (0.455,0.887) |
| Cluster level factors | | |
| **Residence** (ref:urban) | | |
| Rural | 2.902 | (2.316,3.636) |
| **Province**(ref:Cabo Delgado) | | |
| Niassa | 0.743 | (0.506,1.090) |
| Nampula | 1.011 | (0.714,1.429) |
| Zamboza | 0.808 | (0.579,1.130) |
| Tete | 0.373 | (0.243,0.572) |
| Manica | 0.833 | (0.587,1.182) |
| Sofala | 0.605 | (0.421,0.870) |
| Inhambane | 1.277 | (0.862,1.891) |
| Gaza | 0.138 | (0.084,0.227) |
| Maputo Province | 0.021 | (0.007,0.060) |
| Maputo City | 0.032 | (0.007,0.143) |
| **ITN coverage** | 0.25 | (0.110,0.567) |
| **Malaria incidence** | 4.005 | (1.765,9.088) |
| **Spatial Covariance Prams** | | |
| $\sigma^2$ | 0.697 | 0.396 |
| $\phi$ | 0.803 | 0.564 |
| $\tau^2$ | 0.508 | 1.164 |

AOR stands for adjusted odds ratio, and CI for confidence interval.

The likelihood of being parasitaemia for children living with middle and better wealth index household were less likely compared with children living with poor(est) wealth index. Child sex and the availability of bed nets in the household were not significantly related with malaria prevalence ($P-value > 0.05$).

Compared with the non-spatial variation ($\tau^2 = 0.508$) the spatial variation is higher ($\sigma^2 = 0.697$). In the standard non-spatial multilevel modeling, the variance of the random intercept which corresponds to the cluster-level variability is 0.844. Using the non-spatial model the intraclass correlation is 0.204, ($\frac{0.844}{0.844+3.29} = 0.204$), implying that two subjects located in the same cluster had a correlation equal to 0.204 to be parasitaemia.

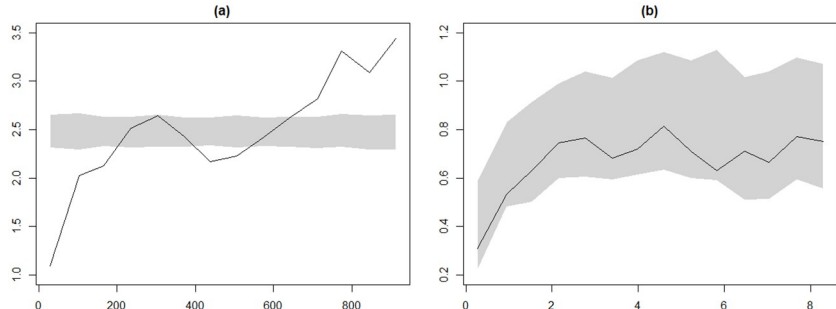

**Fig 3. Plots from variogram diagnostic check for the presence of residual spatial correlation (left-hand panel) and for compatibility of the data with the fitted geostatistical model 3 (right-hand panel).** The solid line is the empirical variogram of the data. The shaded areas are 95% tolerance bands under the hypothesis of spatial independence (left-hand panel) and under the fitted model 3, (right-hand panel).

Fig 4 presents spatial predictions over the study locations, fixing the model parameters at the MCML estimates without any covariate. It also provides the predictive distribution of prevalence in each grid cell through the marginal prevalence (left panel) and probability that the estimated prevalence is above 20% (right panel). Areas with greater than 80% probability of exceeding the threshold were considered hot spots. Central and Northern provinces of Mozambique have locations with predicted prevalence above 20%. The dark green areas show locations where prevalence is above 20%, at 80% certainty.

## Discussion

This study undertook geostatistical analysis of the 2018 MMIS data to identify determinant factors of malaria risk. It also produced contemporary malaria risk maps of Mozambique for children under five years of age across survey clusters and regions of Mozambique. The generated spatially referenced malaria risk map is the first of its kind for Mozambique from a nationally representative geographically-referenced malaria indicator survey data. The map produced illustrates an important synopsis of prevalence of malaria in the country. Therefore, the observed predicted maps can serve as a resourceful tool in planning interventions and a

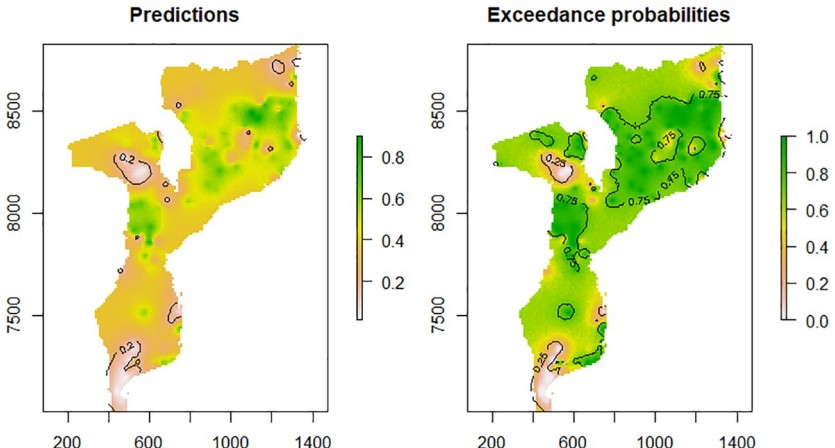

**Fig 4. Malaria prevalence predictions among children aged under five year in Mozambique (left panel) and exceedance probabilities (right panel) for the MMIS data.**

reference point in evaluating their impact across different administrative regions of the country. The predicted map (Fig 4) show that, the health impact of malaria is higher in the northern and lowest in the southern part of the country. This Fig shows that most areas in the northern part of the country are well above a threshold of 20% prevalence. This implies that in these areas, control efforts towards malaria elimination can be considered. For malaria eradication programme implementers, control efforts in these areas would be on reducing transmission through preventive interventions such as mass bed-net distribution and/or indoor residual spraying campaigns. Thus, in the identified high transmission areas, control efforts would need to be more targeted and tailor-made as opposed to universal coverage effort, in order to cut transmission as much as possible. On the other hand, a number of localities in southern part of the country have prevalence below 20% which require less resource to eradicate malaria s compared with the northern part of the country. The following regions: Tete, Gaza, Maputo and Maputo city have locations where predicted malaria prevalence in children under 5 years is less than 20%.

Malaria risk maps generated by [4, 25] relied on historical survey data and do not reflect current malaria prevalence situation for under-five year children in Mozambique. Further, the analyses done by [5, 7, 22, 23] focus on mapping the prevalence of malaria in a certain specific province which does not represent national burden of malaria risk in children under-five years of age. The study conducted by [42] does not represent the contemporary situation of malaria in Mozambique.

As presented in other similar studies conducted to analyze malaria indicator survey datasets [14, 15, 18, 19] the likelihood of having malaria increased as a child gets older. This positive relationship between malaria and child age showed that, the older the child the higher chance to be infected by malaria. This may be due to declining breast feeding which exposes a child to maternal antibodies.

The 95% CI for estimated adjusted odds-ratio obtained using the geostatistical model (3) was narrow compared to results obtained from weighted multilevel model (see Table 3, S1 Table in S1 File). Furthermore, educational level of the mother, sleeping under bednet, and place of residence were found significant in the geostatistical model but not in the non-spatial multilevel analysis (S1 File).

We observed that prevalence estimates vary across different socio-demographic groups as well as different regions of the country. The highest prevalence was observed in Cabo Delgado (57.3% with 95% CI: 51.9-62.5%), Niassa (48.3% with 95% CI: 42.7-53.9%), and Nampula (47.8% with 95% CI: 42.8-52.9%) provinces of the country. This may be due to the fact that, the most populous provinces Zambezia and Nampula have the worst education and health outcomes; and in general northern provinces have worse infrastructure; higher levels of environmental degradation and less economic activity than the south [43]. Further, a secondary school access is unevenly distributed among the provinces and lowers in Cabo Delgado and Niassa province.

The lowest prevalence was observed in Maputo province and Maputo city (Table 1). Children living in urban areas had significantly lower risk of having malaria compared with children in rural areas. Children who live in rural areas were 2.9 times more likely to have malaria than those who live in urban areas (adjusted odds ratio (AOR) = 2.902). Similar results were also found in other studies [13, 18, 44]. Among many other factors, malaria plays a major causative role of anaemia globally [11, 45–48]. In previous studies, the overall prevalence of anaemia in under-five children in Mozambique was above 70% [42]. In this study, we found that anaemic children were found 3.5 times higher to be parasitaemia than non-anemic children.

The positive relationship with age indicated that the older the child the higher the risk of contacting malaria. This findings agrees with results reported from analyses of MIS data in

Nigeria [11], Angola [12], Tanzania([13], Burkina Faso [15], The Gambia [35], Cote d'Ivoire [44], Malawi [20] and Uganda [18].

Recently, Amoah and his colleagues [49] studied the impact of malaria on child growth using 20 Demographic and Health Surveys conducted in 13 African countries. Their result reveals' that malaria had a significant negative effect on child growth.

In line with other similar studies [11–13, 17, 18, 35], this study findings showed that malaria prevalence is strongly associated with mothers' education level, child age, wealth index, cluster level malaria incidence rate, and place of residence. Household wealth index and mothers' educational level were negatively associated with the prevalence of malaria, suggesting that the higher the wealth index and the higher mothers' educational level the lower the risk of acquiring malaria. Supporting this study findings, [12, 13] and [18] found a decreased risk for malaria among children living with better household wealth index, and higher mothers education.

The results presented in this study should be considered in light of some limitations. Since the analysis result in this study derived from a nationwide cross-sectional malaria indicator survey, sub-national variations in risk and epidemiological transitions could be triangulated with additional routine data from health information systems and malaria hospitalization. Further, employing model-based geostatistical methods to interpolate information on malaria prevalence at province/locality levels is less perfect when compared to complete, reliable routine data on the monthly presentation of parasitologically diagnosed fevers to health facilities. Since malaria is environmentally mediated infectious disease, in future studies, considering environmental factors [21, 50] in the covariance structure of the model will yield better prediction map.

## Conclusion

Among other health problems, malaria remains one of the biggest public health problems in Mozambique. Thus, evidence-based interventions are needed to reduce the economic burden [51–53], and malaria related diseases in the country. In this respect, the results of the present study are useful to make geographically targeted interventions.

The results of study showed that high spatial variation in malaria risk were observed across provinces with higher prevalence in the northern part and lower in the southern part of the country. Children living in urban areas had the lowest risk of infection compared with children living in rural areas indicating that more efforts is needed in those areas. Furthermore, the analysis result revealed that malaria risk is linked with child age, household wealth index, mother's educational level, place of residence and child's anaemia level. Moreover, household level determinants of malaria infection that are identified by malaria prevalence maps at cluster and province level could be used in malaria control implementing programs to identify priority intervention areas.

## Supporting information

**S1 File.**
(PDF)

## Acknowledgments

The author thanks Professor Eshetu Wencheko, Dr. Birhanu Teshome, and anonymous reviewers for their valuable comments. The paper has been considerably strengthened by their comments.

## Author Contributions

**Conceptualization:** Bedilu Alamirie Ejigu.

**Data curation:** Bedilu Alamirie Ejigu.

**Formal analysis:** Bedilu Alamirie Ejigu.

**Writing – original draft:** Bedilu Alamirie Ejigu.

**Writing – review & editing:** Bedilu Alamirie Ejigu.

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
