## [Decision Letter · Decision Letter 0]

11 Aug 2020

PONE-D-20-18063

Geostatistical Analysis and Mapping of Malaria Risk in Children of Mozambique

PLOS ONE

Dear Dr.  Ejigu,

Thank you for submitting your manuscript for review to PLoS ONE. After careful consideration, we feel that your manuscript will likely be suitable for publication if it is revised to address the relevant points raised by the reviewer. A significant number of topics need to be clarified and manuscript should be adjusted as suggested.   A Major concern was related to data analysis that should be revised as requested.  For your guidance, a copy of the reviewer' comments was included below. 

We look forward to receiving your revised manuscript.

Kind regards,

Luzia Helena Carvalho, Ph.D.

Academic Editor

PLOS ONE

Journal Requirements:

Reviewers' comments:

Reviewer's Responses to Questions

**Comments to the Author**

1. Is the manuscript technically sound, and do the data support the conclusions?

Reviewer #1: Yes

2. Has the statistical analysis been performed appropriately and rigorously? 

Reviewer #1: Yes

3. Have the authors made all data underlying the findings in their manuscript fully available?

Reviewer #1: Yes

4. Is the manuscript presented in an intelligible fashion and written in standard English?

Reviewer #1: Yes

5. Review Comments to the Author

Reviewer #1: I would like to commend the authors for very good and detailed analysis. A wealth of information has been presented in this paper including a comparison between neighbouring countries. All my comments are quite straight forward

Major comments

1. I have noticed that figure 4 is not adequately described especially in the discussion. It would be great to pinpoint the areas with probabilities above the threshold and explain why that is the case, i.e. link the map with actual locations. The authors should be bold enough in the discussion section to speculate why some areas have high observed probabilities, what are the characteristics of those areas etc. On the same note, it is not clear how the 20% threshold was chosen. Is it a government of Mozambique or WHO standard? This must be made clear.

2. Two major modelling methods are presented here. I thought that the non-spatial approach was done to select the variables to go into the geostatistical model. Table 3 gives the impression that the intention was to compare the 2 approaches. The model-fitting approach should be clearly spelt out in the methods section to avoid any confusion.

3. Varying number of decimal points are presented throughout the paper – sometimes 2 and in most places 3 decimal places. In Table 2, we have 1 decimal place. This must be standardized

4. Discussion section – Are there any limitations associated with this study? You make no mention of any possible limitations. Related to this point, you could also layout future areas of research to address any limitations

Minor comments

1. Line 43 – the word “doubling” is very specific and sounds as if it’s based on empirical evidence. We are simply not sure about the effects of covid on malaria. Instead of doubling, I suggest use a less specific term such as “…increase in the number of malaria deaths…”

2. Line 44 – This line “…affected countries like Mozambique could identify malaria hot-spot areas and move…” the word “could ” in this sentence seems misplaced and as such, sentence is not very clear

3. Section 2.1 – it is okay that you have referenced the reader to the survey final report for details on the methods. However, I feel at least a paragraph summarizing the methods should be included before referring the reader to the report for the rest of the details. Since in this paper you have fitted multilevel models that take into account the clustering, a short description would be nice. You can limit this description to the part of the sampling methodology that introduces the clusters.

4. Line 137, use lower case for “Model”

5. Table 1 – Is it Zamboza or Zambezia province? Please check

6. Table 1 – Until this table, there is no mention of MIS 2011 in the previous section. Shouldn’t it mention somewhere in the methods that there will be comparison between the 2 time points and also comparison between neighbouring countries?

7. Table 1: Mother education label – remove the word “label”

8. Table 1: HH Mosequto bednet – wrong spelling for mosquito

9. Table 1: Child age in year – make is concise – something like “child age (years)”

10. Line 196 - … geostatistical modeling part an exponential function was assumed - Not clear what the exponential function was assumed for. Is for the correlation function. Earlier, you mentioned that a matern function was assumed. Could you please clarify?

11. Line 197 - The assumption provide to be correct as the correlation – I guess it should be “… the assumption proved to be correct …”

12. Line 204 – The sentence “The likelihood of being parasitaemia for children living with household wealth index

13. 204” does not sound very well. I suspect you are missing a word.

14. Table 3 – swap the columns so that you start with non-spatial model (3) and then spatial model (4)

15. From line 244 in the discussion – the paragraph mentions 2 provinces with high prevalence and you mention that prevalence varies across socio-demographic groups. Can you speculate further as to why these 2 provinces have the highest prevalence? What is their socio-demographic profile in comparison with the others? This will make the discussion richer.

6. PLOS authors have the option to publish the peer review history of their article (what does this mean?). If published, this will include your full peer review and any attached files.

Reviewer #1: No

---

## [Author Response · Author response to Decision Letter 0]

17 Sep 2020

Review Comments to the Author

Reviewer #1: I would like to commend the authors for very good and detailed analysis. A wealth of information has been presented in this paper including a comparison between neighboring countries. All my comments are quite straight forward

Major comments

1. I have noticed that figure 4 is not adequately described especially in the discussion. It would be great to pinpoint the areas with probabilities above the threshold and explain why that is the case, i.e. link the map with actual locations. The authors should be bold enough in the discussion section to speculate why some areas have high observed probabilities, what are the characteristics of those areas etc. On the same note, it is not clear how the 20% threshold was chosen. Is it a government of Mozambique or WHO standard? This must be made clear.

- In the revised version of the manuscript, additional description was provided in the result and discussion section which considerably strength the paper.

- In order to guide model-based geostatistical predictions, thresholds of risks were used that resulted in 20% of the population being included in the hotspot based on the theoretical 80–20 assumption where 20% of the population constitutes 80% of the exposure and transmission events (Woolhouse et al 1997, Stresman et al 2017). This threshold selection is likely to be location specific and the hotspot sizes will vary based on the threshold selected: a high threshold would result in only those areas with the highest transmission being identified as a hotspot and a more granular map whereas a less stringent threshold would mean that hotspots would be more ubiquitous. As implemented by different authors (Stresman et al 2017, Chipeta et al 2019,Yankson et al 2019), it common approach to consider 20% as a threshold in the analysis of malaria data.

2. Two major modelling methods are presented here. I thought that the non-spatial approach was done to select the variables to go into the geostatistical model. Table 3 gives the impression that the intention was to compare the 2 approaches. The model-fitting approach should be clearly spelt out in the methods section to avoid any confusion.

- As you noticed, non-spatial multilevel model were fitted to identify significant variables by taking into account the sampling weight. Since the focus in this study is to analyze 2018 MMIS using model based geostatistics, to avoid confusion I presented results only obtained using geostatistical model. As a result, descriptions of multilevel modeling from the methods section and parameter estimates from the result table were dropped. 

3. Varying number of decimal points are presented throughout the paper – sometimes 2 and in most places 3 decimal places. In Table 2, we have 1 decimal place. This must be standardized

- This issue addressed throughout the revised manuscript. 

4. Discussion section – Are there any limitations associated with this study? You make no mention of any possible limitations. Related to this point, you could also layout future areas of research to address any limitations.

- Additional paragraph mentioning limitations associated with this study included in the revised version.

Minor comments

1. Line 43 – the word “doubling” is very specific and sounds as if it’s based on empirical evidence. We are simply not sure about the effects of covid on malaria. Instead of doubling, I suggest use a less specific term such as “…increase in the number of malaria deaths…”

 - Corrected.

2. Line 44 – This line “…affected countries like Mozambique could identify malaria hot-spot areas and move…” the word “could ” in this sentence seems misplaced and as such, sentence is not very clear

 -Corrected.

3. Section 2.1 – it is okay that you have referenced the reader to the survey final report for details on the methods. However, I feel at least a paragraph summarizing the methods should be included before referring the reader to the report for the rest of the details. Since in this paper you have fitted multilevel models that take into account the clustering, a short description would be nice. You can limit this description to the part of the sampling methodology that introduces the clusters.

 - Additional description about survey methodology included.

4. Line 137, use lower case for “Model”

 - Corrected.

5. Table 1 – Is it Zamboza or Zambezia province? Please check

- Corrected as Zambezia. 

6. Table 1 – Until this table, there is no mention of MIS 2011 in the previous section. Shouldn’t it mention somewhere in the methods that there will be comparison between the 2 time points and also comparison between neighbouring countries?

 - A short description included in the methods Section. 

7. Table 1: Mother education label – remove the word “label”

 -Corrected. 

8. Table 1: HH Mosequto bednet – wrong spelling for mosquito

 -Corrected.

9. Table 1: Child age in year – make is concise – something like “child age (years)”

 -Corrected.

10. Line 196 - … geostatistical modeling part an exponential function was assumed - Not clear what the exponential function was assumed for. Is for the correlation function. Earlier, you mentioned that a matern function was assumed. Could you please clarify?

-Yes, it is for the correlation function, and corrected accordingly. Exponential correlation function is a special case of Matern function when k=1/2.

11. Line 197 - The assumption provide to be correct as the correlation – I guess it should be “… the assumption proved to be correct …”

- Corrected.

12. Line 204 – The sentence “The likelihood of being parasitaemia for children living with household wealth index” does not sound very well. I suspect you are missing a word.

-Yes, corrected. 

14. Table 3 – swap the columns so that you start with non-spatial model (3) and then spatial model (4)

- As stated earlier, to avoid confusion, results from multilevel model (3) dropped.

15. From line 244 in the discussion – the paragraph mentions 2 provinces with high prevalence and you mention that prevalence varies across socio-demographic groups. Can you speculate further as to why these 2 provinces have the highest prevalence? What is their socio-demographic profile in comparison with the others? This will make the discussion richer.

- Thanks. The following statement included in the revised version.

“…..This may be due to the fact that, the most populous provinces Zambezia and Nampula have the worst education and health outcomes; and in general northern provinces have worse infrastructure; higher levels of environmental degradation and less economic activity than the south \\cite{SIDA, 2019}. Further, a secondary school access is unevenly distributed among the provinces and lowers in Cabo Delgado and Niassa province (Fox et al, 2019). “

---

## [Decision Letter · Decision Letter 1]

5 Oct 2020

PONE-D-20-18063R1

Geostatistical Analysis and Mapping of Malaria Risk in Children of Mozambique

PLOS ONE

Dear Dr. Ejigu,

Thank you for submitting your manuscript for review to PLoS ONE. After careful consideration, we feel that your manuscript will likely be suitable for publication if the authors revise it to address additional points raised by the reviewers.  According to reviewers, there are some specific areas where further improvements would be of substantial benefit to the readers, including supplementary material. Finally, the MS should be submitted to a copy-editing process otherwise the readability of the MS may be  compromised. For your guidance, a copy of the reviewers' comments was included below.

We look forward to receiving your revised manuscript.

Kind regards,

Luzia Helena Carvalho, Ph.D.

Academic Editor

PLOS ONE

Reviewers' comments:

Reviewer's Responses to Questions

**Comments to the Author**

1. If the authors have adequately addressed your comments raised in a previous round of review and you feel that this manuscript is now acceptable for publication, you may indicate that here to bypass the “Comments to the Author” section, enter your conflict of interest statement in the “Confidential to Editor” section, and submit your "Accept" recommendation.

Reviewer #1: All comments have been addressed

Reviewer #2: All comments have been addressed

2. Is the manuscript technically sound, and do the data support the conclusions?

Reviewer #1: Yes

Reviewer #2: Yes

3. Has the statistical analysis been performed appropriately and rigorously? 

Reviewer #1: Yes

Reviewer #2: Yes

4. Have the authors made all data underlying the findings in their manuscript fully available?

Reviewer #1: Yes

Reviewer #2: Yes

5. Is the manuscript presented in an intelligible fashion and written in standard English?

Reviewer #1: Yes

Reviewer #2: Yes

6. Review Comments to the Author

Reviewer #1: Since the non-spatial model results have been removed from the manuscript in response to my earlier comment, I suggest that the authors consider including them as part of supplementary material. In this way, interested readers can go over to the supplementary material to get more details

Reviewer #2: Even though the writer of this manuscript is sole author, while writing it, he frequently said "our understanding, our knowledge". Is that appropriate being a sole author?

7. PLOS authors have the option to publish the peer review history of their article (what does this mean?). If published, this will include your full peer review and any attached files.

Reviewer #1: No

Reviewer #2: No

---

## [Author Response · Author response to Decision Letter 1]

16 Oct 2020

Review Comments to the Author

Reviewer #1: Since the non-spatial model results have been removed from the manuscript in response to my earlier comment, I suggest that the authors consider including them as part of supplementary material. In this way, interested readers can go over to the supplementary material to get more details.

- Model description and results obtained from the non-spatial model have been submitted as supplementary material. 

Reviewer #2: Even though the writer of this manuscript is sole author, while writing it, he frequently said "our understanding, our knowledge". Is that appropriate being a sole author?

- Thanks a lot for your valuable comment. Addressed.

---

## [Editor Report · Decision Letter 2]

20 Oct 2020

Geostatistical Analysis and Mapping of Malaria Risk in Children of Mozambique

PONE-D-20-18063R2

Dear Dr. Ejigu,

We’re pleased to inform you that your manuscript has been judged scientifically suitable for publication and will be formally accepted for publication once it meets all outstanding technical requirements.

Kind regards,

Luzia Helena Carvalho, Ph.D.

Academic Editor

PLOS ONE
---

## [Editor Report · Acceptance letter]

23 Oct 2020

PONE-D-20-18063R2 

Geostatistical Analysis and Mapping of Malaria Risk in Children of Mozambique 

Dear Dr. Ejigu:

I'm pleased to inform you that your manuscript has been deemed suitable for publication in PLOS ONE. Congratulations! Your manuscript is now with our production department. 

Kind regards, 

on behalf of

Dr. Luzia Helena Carvalho 

Academic Editor

PLOS ONE